# Fabrication of Diatomite/Silicalite-1 Composites and Their Property for VOCs Adsorption

**DOI:** 10.3390/ma12040551

**Published:** 2019-02-13

**Authors:** Yutong Liu, Tao Tian

**Affiliations:** Key Laboratory of Groundwater Resources and Environment, Ministry of Education, College of New Energy and Environment, Jilin University, Changchun 130012, China; liuyt841011@163.com

**Keywords:** Diatomite, silicalite-1, hierarchical porous structure, adsorption, VOCs

## Abstract

Adsorption technology is an effective method to remove volatile organic compounds (VOCs). In this work, we prepared hierarchical porous materials using modified diatomite (Dt) as a support and nano-sized silicalite-1 (S-1) seeds as inorganic fillers, which were applied to adsorb volatile organic compounds (VOCs). The characterization of the composites indicated that S-1 was successfully coated onto the surface of modified Dt, and the best surface area of the composites was 398.8 m^2^/g, nearly 40 times as large as Dt. The adsorption capacities of Dt/S-1 composites for three probe VOCs (ethyl acetate, acetone, and toluene) were rather superior to Dt, and the composites had preferential adsorption selectivity for ethyl acetate. Effects of seeded zeolite contents and hydrothermal conditions for the adsorption capacity of composites were discussed in this paper. The composite seeded with 5 wt% S-1 zeolite, which was subsequently synthesized by hydrothermal reaction at 100 °C for four days, showed the maximum adsorption capacity (1.31 mmol/g for ethyl acetate). The pseudo second-order model provided a perfect fit to adsorption kinetics, while the Langmuir model agreed the best with the adsorption isotherms. In addition, the composites had selective adsorption to ethyl acetate among these three probes VOCs. The regeneration experiments were also carried out, and the adsorption efficiency of the adsorbents was still up to 67% after five adsorption–desorption cycles. The hierarchical porous Dt/S-1 composites have an excellent VOC adsorption performance, satisfactory selectivity, and recycling ability.

## 1. Introduction

Due to fast urbanization and industrialization, the emissions of volatile organic compounds (VOCs) have increased dramatically over the last decade and have received considerable attention because of the severe environmental hazards they create [1]. The term VOCs refers to organic compounds that have a saturated vapor pressure greater than 133.3 Pa at room temperature and a boiling point that varies from 50 °C to 260 °C [2]. VOCs discharged into the atmosphere could become precursors of ozone and secondary organic aerosols by taking part in photochemical reactions with nitrogen oxides under sunlight. The harmful products could furthermore induce photochemical smog and haze and cause serious implications for human health and activities [3].

A number of studies have indicated the effectiveness of VOC removal by adsorption technology. The most frequently used materials for the capture of VOCs are porous substances with large surface areas and pore volume such as zeolites [4], resins [5] and activated carbons together with their derivatives [6,7,8,9]. It has been demonstrated that zeolites are desirable materials for the removal of hydrocarbons because of their “tailor-made” physico-chemical properties [10], together with their high thermal and chemical stability and renewable ability [11]. However, most zeolites have a strong hydrophilic function, resulting in water molecules taking up the adsorption sites of organic molecules. It has been found that pure silica zeolites tend to be hydrophobic, such as nano-sized silicalite-1 (S-1) of MFI-type zeolite [12]. Nevertheless, two major issues still exist in the application of nano S-1 as adsorbents. On the one hand, the cellular structure of zeolite leads to their adsorption to only a certain size of small organic molecules, which is difficult to meet the actual needs of industries. On the other hand, nanoparticle agglomeration might reduce the effective surface area and lower the adsorption capacity [13,14,15].

Diatomite (Dt) is a natural siliceous rock composed of microfossils of aquatic algae called diatoms [16,17]. Dt is abundant in China with low cost, a unique macroporous structure (50–800 nm), and high thermal stability [18]. However, raw Dt exhibits a relatively low adsorption capacity due to its poor surface area. Proper modification would significantly improve the surface properties of raw Dt, considering that raw Dt could be a suitable carrier material [19]. Anderson et al. [20] synthesized a composite by loading S-1 seeds onto Dt. The obtained composite showed a low specific surface area (*S_BET_*, 29.2 m^2^/g) and micropore volume (*V_micropore_*, 0.01 cm^3^/g). Yuan et al. [21] reported that a hierarchically porous Dt/S-1 composite for benzene adsorption was fabricated via a facile pre-modification in situ synthesis route. The surface area and micropore volume of the composite were improved a lot, however, in situ synthesis without crystal seeds has its limitations such as uneven load and active components easily draining away. Furthermore, as for the application of Dt/S-1 on VOC adsorption, it is of great significance to explore multi-component and competitive adsorption of VOCs.

In this study, the highly dispersed nano S-1 modified Dt (Dt/S-1) for VOC adsorption was synthesized by a pre-modification and two-step crystallization method. This kind of method could make zeolites grow regularly around S-1 seeds on the surface of Dt, which may avoid zeolites blocking up large pores of Dt at the same time. The adsorbents were characterized by X-ray diffraction (XRD), scanning electron microscopy (SEM), and Brunauer–Emmett–Teller (BET) analysis. Different preparation conditions such as doping ratio, hydrothermal temperature and hydrothermal time were investigated. Acetone, ethyl acetate, and toluene were used as probe adsorbates to systematically evaluate the adsorption performance of the obtained Dt/S-1 composite. Single-component and multi-component adsorption experiments were exhibited as a comparison. Adsorption kinetics, isotherms and selectivity were included as adsorption assessments. Additionally, the regeneration of the adsorbents was also discussed in this paper.

## 2. Materials and Methods 

### 2.1. Chemicals

The synthesis of S-1 zeolite seeds was conducted from a mother solution containing tetraethylorthosilicate, (TEOS, 98%, Alfa Aesar), tetrapropylammonium hydroxide (TPAOH, 20 wt% in water, Alfa Aesar) and ultrapure water. Raw Dt powders were obtained from the Changbai deposit in Jilin Province, China. Hydrochloric acid (Beijing Chemical Plant), sodium hydrate (NaOH, Beijing Chemical Plant), and polydiallyldimethylammonium chloride (PDDA, 20 wt% in water, Alfa Aesar) were used to modify Dt. Three organic compounds were selected as the probe adsorbates. Acetone (CH_3_COCH_3_), ethyl acetate (CH_3_COOC_2_H_5_), and toluene (CH_3_C_6_H_5_) were purchased from Beijing Chemical Plant.

### 2.2. Preparation of Dt/S-1 Composites

#### 2.2.1. Synthesis of S-1 Zeolite Seeds

Nano-sized S-1 zeolite was synthesized from a colloidal precursor solution with the following chemical compositions: 25 SiO_2_:9 TPAOH:480 H_2_O. The silica source for the preparation of the initial precursors was TEOS, and the alkali source was TPAOH. These components were mixed under vigorous stirring and aged on an orbital shaker at ambient temperature for 12 h prior to the further hydrothermal treatment at 100 °C for 4 days. After the hydrothermal synthesis, the seeds were washed 3 times by repetitive dispersions in water applying a 5-min ultrasonic treatment followed by a 10 min centrifugation at 8000 rpm. Finally, the seeds were dried at 60 °C before being calcined at 550 °C for 4 h to remove the organic template.

#### 2.2.2. Pre-Modification of Dt

Firstly, Dt was mixed with 6 mol/L hydrochloric acid solution to remove impurities [22]. After the mixture was stirred vigorously for 3 h in a thermostatic bath pot at 60 °C, Dt was washed with deionized water until the pH reached 7–8 and dried in a ventilated oven at 60 °C.

Secondly, the Dt treated by acid was further modified with alkali to enlarge the pores to prevent zeolites blocking up the pores. The Dt was mixed with sodium hydroxide solution (pH = 13.5). The mixture was stirred vigorously for 3 h in a thermostatic bath pot at 60 °C. Then the Dt was washed until the pH reached 7–8 and dried at 60 °C after separation.

Finally, 1 g Dt treated as above was mixed with 20mL of 0.5 wt% PDDA solution. PDDA modification could change the surface charge of Dt from negative to positive [23], which was beneficial for combination with negatively charged S-1 seeds resulting from the electrostatic attraction. Then the mixture was stirred for 1 h and aged for 30 min before it was washed and dried at 60 °C.

#### 2.2.3. Seeds-Assisted Synthesis of Dt/S-1 Composites

A certain amount of nano-sized S-1 zeolite was dispersed into an ammonia solution (pH = 9.5), then 1 g modified Dt was added into it. The mixture was stirred for 1 h and aged for 30 min, then washed 3 times with 0.1 mol/L ammonia solution to finish the seeding procedure. Additionally, a certain amount of the seeded sample was mixed with 40 mL zeolite colloidal precursor solution (reported in Section 2.2.1) for further hydrothermal synthesis at 100 °C for 4 days. The obtained mixture was washed repeatedly with water, dried at 60 °C, and finally calcined at 550 °C for 4 h. In this paper, the effect of seeded zeolite content (1 wt%, 5 wt%, 10 wt% and 20 wt%), hydrothermal temperature (90 °C, 100 °C and 110 °C) and hydrothermal time (3 days, 4 days and 5 days) were investigated. 

### 2.3. Characterization of Adsorbents

X-ray diffraction (XRD) patterns were recorded by an X-ray diffractometer (Bruker D8 ADVANCE, Karlsruhe, German) with a Cu Kα radiation source (k = 0.154 nm) operated under a generating voltage of 40 kV and a current of 40 mA. Scanning electron microscopy (SEM) images were obtained with a scanning electron microscope (TOPCON ABT-150S, Tokyo, Japan). A Micromeritics ASAP 2020 system was used to measure the N_2_ adsorption–desorption isotherms (N_2_ adsorption isotherms at 77 K, and all samples were pre-activated at 300 °C under vacuum for 10 h). The surface area was calculated from the N_2_ adsorption data using the multi-point Brunauer–Emmett–Teller (BET) equation [24]. The micropore volume was obtained via the t-plot method.

### 2.4. Adsorption Experiments

Adsorption experiments were performed in a 125 mL container with 0.01 g adsorbents. A specific amount of liquid probe VOC (acetone, ethyl acetate, and toluene) was injected into the container before it was sealed. The liquid organics were converted into vapour and adsorbed by the adsorbents at 60 °C. The concentrations of residual organics were determined by a gas chromatograph device (GC-2014C, SHIMADZU, Japan) equipped with a WonderCap5 column and a flame ionization detector (FID). The temperatures of the inlet, analyzer and column were 240 °C, 300 °C and 70 °C, respectively. The equilibrium time was 4 min. Adsorption experiments were performed with an initial vapour concentration of 0.12 mmol/L, except for adsorption isotherms experiments. Kinetics experiments were performed at different times, ranging from 5 min to 60 min, while adsorption isotherms experiments were performed with initial vapour concentrations varying from 0.04 mmol/L to 0.27 mmol/L. Competitive adsorption experiments were conducted with multi-component vapour in a single container with the same initial concentration. To evaluate regeneration capacity, the adsorption–desorption processes of ethyl acetate were performed for 6 cycles, and the desorption experiments were carried out by heating the inactive materials at 120 °C for 6 h. Equilibrium adsorption capacity, *Q_e_*, of the adsorbent was calculated as Equation (1):(1)$$Q_{e}=(C_{0}-C_{e})\cdot V/W$$ where *C_0_* (mmol/L) and *C_e_* (mmol/L) are the initial and equilibrium concentrations of the adsorbates. *V* (L) and *W* (g) represent the vapour volume and the mass of Dt/S-1, respectively.

## 3. Results and Discussion

### 3.1. Characterization of the Adsorbents

#### 3.1.1. XRD

The X-ray diffraction (XRD) patterns are shown in Figure 1. The XRD pattern of Dt revealed the main phase of non-crystalline opal-A with the characteristic broad peak centered at 21.8° [25]. S-1 zeolite samples presented obvious diffraction peaks at about 7.8°, 8.8°, 14.8°, 17.6°, 23°, 23.6° and 24.4°, consistent with the crystal characteristic peaks of S-1 in the standard spectra [26]. Compared with S-1, Dt/S-1 (5 wt%) almost owned the same characteristic diffraction peaks, indicating that S-1 was successfully loaded onto the surface of Dt.

#### 3.1.2. SEM

SEM observations (Figure 2) revealed the surface appearance of Dt (a, b) and Dt/S-1 (5 wt%) (c, d). Dt displayed a disk structure with a diameter between 20 μm to 25 μm. In particular, there were large pores around 0.1–0.5 μm with a regular distribution. Figure 2c,d showed that Dt was successfully loaded by S-1 crystals with the size of 60–70 nm. Macropores were observed on the composites as shown in Figure 2d.

#### 3.1.3. BET

The surface properties of Dt, S-1 and the Dt/S-1 composites determined by N_2_ adsorption–desorption were listed in Table 1 and Appendix A. The surface area of Dt/S-1 (5 wt%) was nearly 40 times as large as Dt. The total pore volume and micropore volume of Dt/S-1 (5 wt%) were both much superior to that of Dt. The considerable porous parameters of the composite were attributed to the nano-sized S-1 coated onto Dt. In addition, the wt% of zeolite in the composites was calculated using Equation (2) [21]:(2)$$W_{zeolite}\%=\left[V_{micropore}\left(composite\right)-V_{micropore}\left(Dt\right)\right]/V_{micropore}\left(S-1\right)\times 100\%$$

According to the equation, the wt% of zeolite in Dt/S-1 (5 wt%) was 74.1%, and the content of coated zeolite was higher than the work reported before.

Figure 3 showed the N_2_ adsorption–desorption isotherms of Dt, S-1 and Dt/S-1 (5 wt%). The isotherm of raw Dt featured a type II curve with a minor H3 hysteresis loop according to the IUPAC classification [27], which indicated that Dt contained small quantities of mesopores. When P/P_0_ ≤ 0.1, Dt had poor adsorption to N_2_, which implies that Dt had few micropores. The rapidly increased adsorption quantities when P/P_0_ ≈ 1.0 suggest abundant macroporosity. S-1 had a type IV adsorption–desorption isotherm curve with an evident H3 hysteresis loop, which was an indication of the formation of mesopores from the nano-crystal stacking. The adsorption of N_2_ on S-1 increased rapidly while P/P_0_ ≤ 0.1 because of the rapid filling of N_2_ into the micropores of zeolites. In addition, the adsorption rose again rapidly when P/P_0_ ≈ 1.0 according to the nano-size effect and particle agglomeration. The Dt/S-1 (5 wt%) had a type IV adsorption–desorption isotherm curve and evident H3 hysteresis loop, and the adsorption of N_2_ increased rapidly while P/P_0_ ≤ 0.1, similar to S-1, indicating that the composite consisted of nano-sized S-1 particles. Finally, the increasing trend of N_2_ adsorption of Dt/S-1 (5 wt%) in the high pressure region (P/P_0_ ≈ 1.0) was moderated compared with that of S-1, which explained that the composite materials still had a considerable number of large pores. Above all, the hierarchical porous structure was synthesized successfully, making up for the disadvantages of pure Dt and zeolite. The other samples had similar N_2_ adsorption–desorption isotherms to Dt/S-1 (5 wt%) which are shown in Appendix A.

### 3.2. VOC Adsorption Capacity Tests

#### 3.2.1. Effects of Seeded Zeolite Contents and Hydrothermal Conditions

The equilibrium adsorption capacities for the adsorption of three VOCs on various adsorbents were exhibited in Table 2. Experimental results showed that the adsorption capacities of the adsorbents with different loadings of S-1 seeds were 4–22 times that of Dt. In addition, with the increase of seed content from 1 wt% to 10 wt%, ethyl acetate adsorption capacities on the adsorbents were improved from 1.10 mmol/g to 1.31 mmol/g, and the specific surface area of the samples increased from 319.2 m^2^/g to 402.3 m^2^/g (Appendix A). However, the adsorption capacities of the three VOCs declined obviously as the seed content increased up to 20 wt%. The specific surface area of Dt/S-1 (20 wt%) decreased to 336.8 m^2^/g. While the micropore volume improved, the total volume had decreased, indicating that the dispersed S-1 nanocrystals were occupied inside the pores of Dt. On the one hand, the hierarchical porous structure of Dt/S-1 enhanced the efficiency of large Dt pores and made up for the shortcomings of pure zeolite for its limited pore size distribution. On the other hand, the dispersion of S-1 on Dt could reduce the agglomeration of nanoparticles and decrease mass transfer resistance. S-1 seeds played a guiding role in the growth of zeolite, but excessive load might cause overgrowth of crystals and lead to particle agglomeration, reducing the surface area of the adsorbents exposed to probe VOCs. The adsorption effect of Dt/S-1 (5 wt%) and Dt/S-1 (10 wt%) were similar to pure S-1. Considering the actual content of zeolite in the composites (74.1% in Dt/S-1 (5 wt%)), the utilization efficiency of zeolite was greatly improved. Moreover, due to the introduction of diatomite, the synthesis cost of the composite material is significantly lower than that of pure S-1 zeolite. 

The adsorption capacities of probe VOCs were ethyl acetate > acetone > toluene. According to existing research, the interaction between adsorbents and adsorbates is mostly the consequence of the comprehensive effect of adsorbents’ pore canal structure and adsorbates’ physicochemical qualities [28,29,30,31,32,33]. S-1 zeolite possessed sinusoidal channels with 0.54 nm circular cross-sections interconnected with straight channels with 0.51 nm × 0.57 nm elliptical cross sections [34]. If the adsorption became effective and stable, the adsorbent’s pore size must be close to the size of the adsorbates [30]. The sizes of acetone and ethyl acetate were in the effective range of adsorption, but toluene was not. Abundant research has indicated that high-silica MFI-type zeolites have a nonpolar nature [12], and thus S-1 zeolite tended to adsorb substances of low polarity. Since the polarity of ethyl acetate was smaller than acetone, according to the like-dissolves-like theory, the composite adsorbents were more likely to adsorb ethyl acetate rather than acetone. 

The equilibrium adsorption capacities of Dt/S-1 (5 wt%) prepared at different hydrothermal temperatures are shown in Table 3. The adsorbent synthesized at 100 °C showed the most considerable adsorption capacity, which was superior to the adsorbents synthesized at 90 °C and 110 °C. With the increase in temperature, the crystallinity was improved accordingly (Appendix A). Nevertheless, higher temperatures would cause the growth of over-sized molecules or the agglomeration of particles, and then had a negative impact on the surface area and pore volume of the adsorbents [35]. 

Table 4 shows the equilibrium adsorption capacities of Dt/S-1 (5 wt%) prepared at different hydrothermal times. The results showed that the adsorbent synthesized for four days had the most desirable adsorption capacity. With the increase in hydrothermal time, the overgrowth of S-1 zeolite might lead to the occlusion of Dt’s large pores [36] and reduce the adsorption efficiency of the adsorbents.

#### 3.2.2. Adsorption Kinetics

Figure 4 demonstrates the kinetic adsorption process of Dt/S-1 (5 wt%) to probe VOCs. It took 25 min, 35 min and 45 min for toluene, acetone and ethyl acetate to reach the adsorption equilibrium, respectively. Experimentally determined kinetic data were analyzed using nonlinear pseudo first-order and pseudo second-order models using Equations (3) and (4) [37]:(3)$$Q_{t}=Q_{e}(1-e^{-k_{1}t})$$
(4)$$Q_{t}=\frac{k_{2}Q_{e}{}^{2}t}{1+k_{2}Q_{e}t}$$
where, *k_1_* (min^−1^) is the pseudo first-order rate constant of adsorption, *k_2_* (g·mol^−1^min^−1^) is the pseudo second-order rate constant of adsorption, *Q_e_* (mol·g^−1^) is the adsorption capacity at equilibrium and *Q_t_* (mol·g^−1^) is the adsorption capacity at a given time *t*.

The obtained kinetic parameters were shown in Table 5. It was noted that *Q_e_* (exp) represents data from experiments, while *Q_e_* (cal) represents data from model fittings. It was observed that the pseudo second-order model well fit the experimental data of the three organic compounds with a higher correlation coefficient (*R*^2^). Also, *Q_e_* (cal) values from the pseudo second-order model were found to agree better to the experimentally obtained *Q_e_* (exp). The fitting results thus indicated that the adsorption rate was mainly determined by the chemical adsorption process [37].

#### 3.2.3. Adsorption Isotherms

Figure 5 demonstrated that the adsorbed amount of probe VOCs on Dt/S-1 (5 wt%) increased with increasing initial concentrations. In order to figure out adsorption isotherms of the adsorbent, the Henry model, Langmuir model and Freundlich model were utilized with Equations (5)–(7) [38]:(5)$$Q_{e}=k_{H}C_{e}$$
(6)$$Q_{e}=\frac{Q_{\max}k_{L}C_{e}}{1+k_{L}C_{e}}$$
(7)$$Q_{e}=k_{F}C_{e}{}^{1/n}$$
where *Q_max_* (mol·g^−1^) is the maximum adsorption capacity, *k_H_* is the Henry constant (L·g^−1^), *k_L_* (L·mmol^−1^) is the Langmuir equilibrium constant, *k_F_* (mmol^1−n^·L^n^·g^−1^) is the Freundlich constant and 1/n is the heterogeneity factor. 

The adsorption isotherm parameters are listed in Table 6. The Langmuir model yields a much better fit than that of the Henry model or the Freundlich model, indicating that the adsorption process happened on a monolayer.

#### 3.2.4. Adsorption Selectivity

In addition to single-component adsorption, this study also explored multi-component adsorption onto Dt/S-1 (5 wt%). Figure 6 presents the competitive adsorption of three probe VOCs with the same initial vapor concentrations. After five minutes, the adsorption value of acetone was close to ethyl acetate, and the adsorption of toluene was far lower than the former two organics. Subsequently, the adsorption of ethyl acetate continued to increase, while the adsorption of acetone decreased sharply. After 45 min, adsorption of the three VOCs reached an equilibrium. In comparison with the adsorption of a single component, the adsorption capacities decreased since adsorption competition existed among the three probe organics. In a comprehensive way, single-component and multi-component adsorption experiments showed that the composite had the capability of selective adsorption to ethyl acetate.

The distribution coefficient (*K_d_*) was used to analyze the selectivity of the absorbent toward three probe VOCs. The equation is stated as [39]:(8)$$k_{d}=\frac{V\cdot (C_{0}-C_{e})}{m\cdot C_{e}}$$ where *C_0_* and *C_e_* (mmol/L) represent the initial and equilibrium concentrations of solutes, respectively; *V* (L) is the volume of solution; and *m* (g) is the mass of the adsorbent.

A selectivity coefficient (*α*) for the binding of a particular adsorbate in the presence of interfering compounds is defined in Equation (9) as [39]:(9)$$\alpha =k_{d}(T)/k_{d}(I)$$ where *K_d_* (T) is the *K_d_* value of the targeted compound (ethyl acetate in this case) and *K_d_* (I) is the *K_d_* value of the other compound in the multi-substance mixtures (acetone or toluene here). A larger value of α indicates greater selectivity toward ethyl acetate than acetone or toluene. The calculated *k_d_* of acetone, ethyl acetate and toluene were 6.25, 15 and 0.47 L/g, and the α values of ethyl acetate to acetone and toluene were 2.4 and 31.91, respectively. These results showed better adsorption capacities for ethyl acetate than acetone or toluene, indicating that Dt/S-1 (5 wt%) had selective adsorption toward ethyl acetate.

#### 3.2.5. Regeneration of the Adsorbents

The regeneration of Dt/S-1 (5 wt%) was carried out by performing five consecutive adsorption–desorption cycles under the same experimental conditions. Figure 7 illustrates the adsorption capacities of Dt/S-1 (5 wt%) at different adsorption–desorption cycles. The adsorption capacities decreased from the first to the fifth cycle from 85.4% to 67%, which could be attributed to the remaining VOCs in the adsorbents during the regeneration process or the morphology change of Dt/S-1 (5 wt%) [40]. After five cycles, the adsorption efficiency was still more than 60%. Therefore, we could conclude that Dt/S-1 (5 wt%) was suitable for its use and reuse with high removal and recovery. Regeneration is the premise of adsorbents recycling. It is of great significance to reduce the operating cost and increase the spread of adsorption technology.

## 4. Conclusions

In this paper, nano S-1 seeds were loaded onto the surface of Dt using the advanced hydrothermal method, which was characterized by XRD, SEM and BET. The Dt/S-1 (5 wt%) exhibited a considerably higher VOC adsorption capacity compared to raw Dt and other composites prepared under different conditions. The pseudo second-order model provided the perfect fit to the dynamic behavior of VOC adsorption onto the composite material for the whole contact time period, while the Langmuir model agreed the best with the adsorption isotherms in terms of different initial concentrations. Further study discovered that the composite had selective adsorption to ethyl acetate among the three VOCs. With a considerable regeneration capacity, the composite material rendered its potential for application in VOC removal techniques.

## Figures and Tables

**Figure 1 materials-12-00551-f001:**
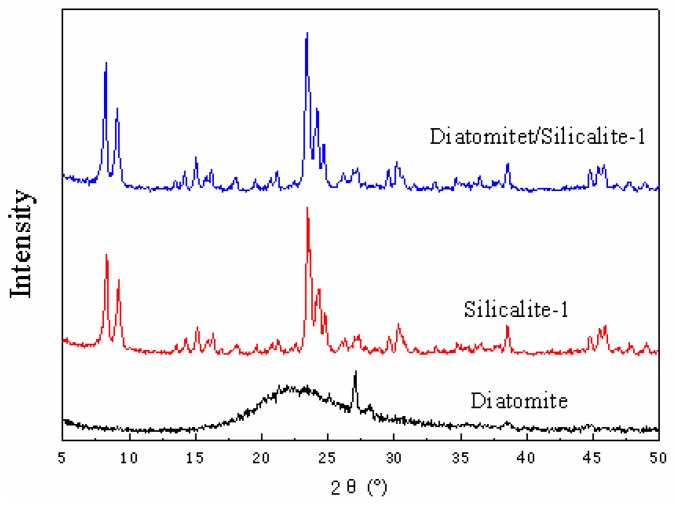
X-ray diffraction (XRD) patterns of diatomite (Dt), silicalite-1 (S-1) and Dt/S-1 (5 wt%).

**Figure 2 materials-12-00551-f002:**
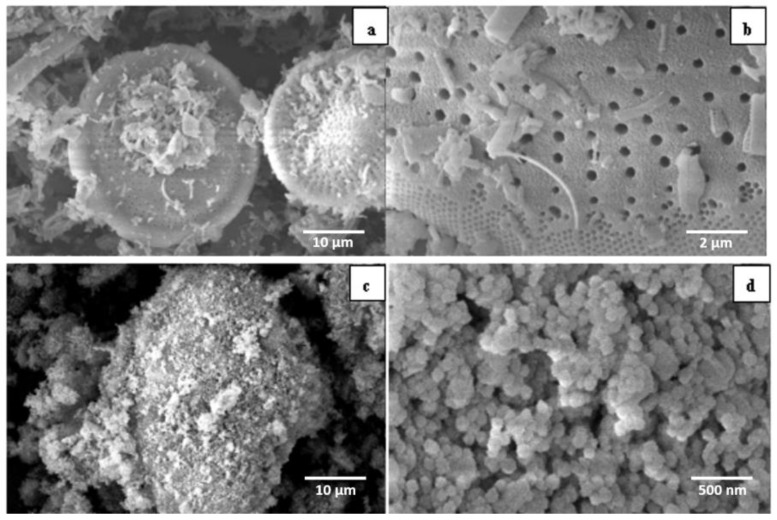
Scanning electron microscopy (SEM) images of Dt (**a**,**b**) and Dt/S-1 (5 wt%) (**c**,**d**).

**Figure 3 materials-12-00551-f003:**
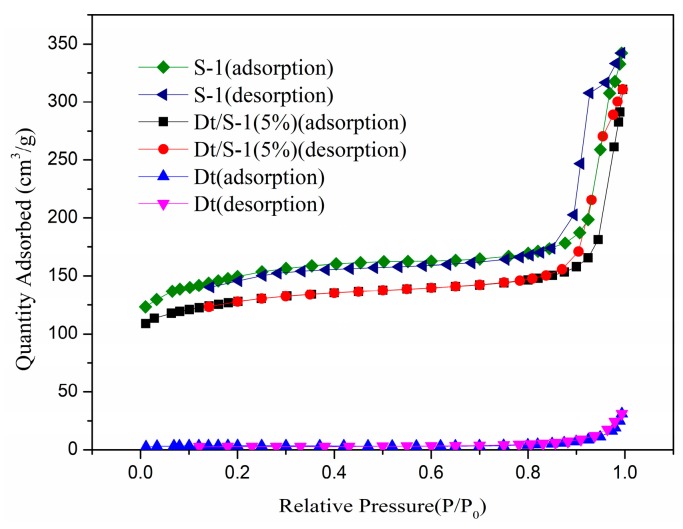
N_2_ adsorption–desorption isotherms of Dt, S-1 and Dt/S-1 (5 wt%).

**Figure 4 materials-12-00551-f004:**
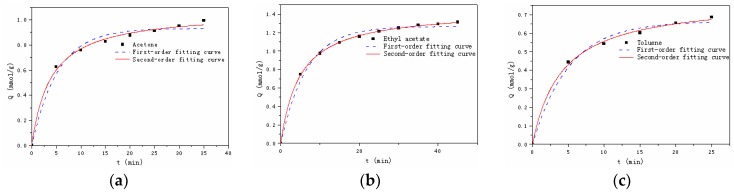
Kinetic adsorption process of Dt/S-1 (5 wt%) to acetate (**a**), ethyl acetate (**b**) and toluene (**c**).

**Figure 5 materials-12-00551-f005:**
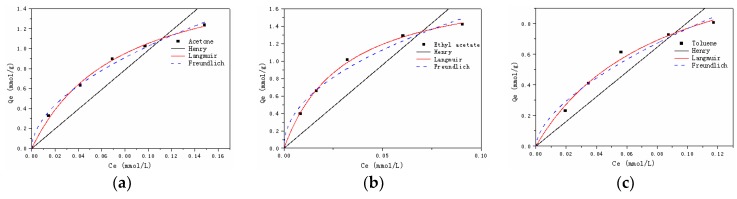
Equilibrium adsorption capacities of acetate (**a**), ethyl acetate (**b**) and toluene (**c**) at different equilibrium concentrations.

**Figure 6 materials-12-00551-f006:**
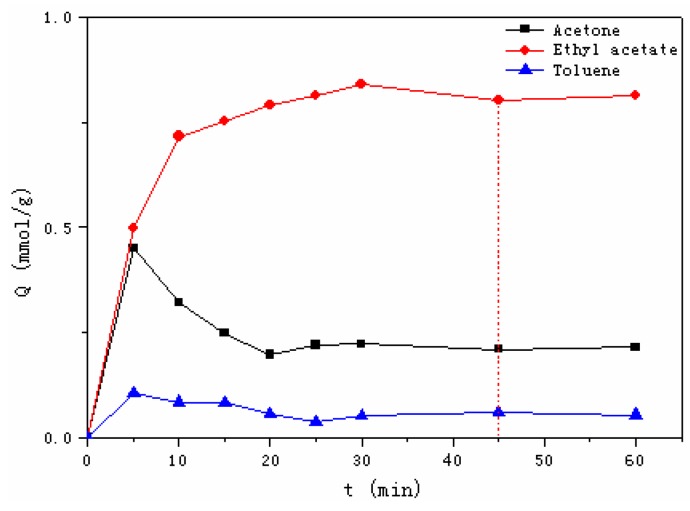
Competitive adsorption of acetate, ethyl acetate and toluene.

**Figure 7 materials-12-00551-f007:**
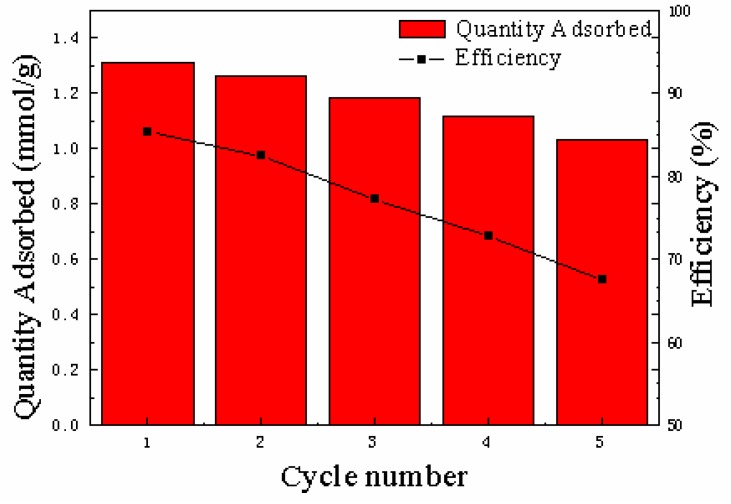
Ethyl acetate adsorption capacities (mmol/g) and efficiency (%) for the use of Dt/S-1 (5 wt%) at different adsorption–desorption cycles (up to five).

**Table 1 materials-12-00551-t001:** Porous parameters of Dt, S-1 and Dt/S-1 (5 wt%). *S_BET_* = specific surface area. *V_micropore_* = micropore volume. *V_total_* = total volume.

Sample	Dt	S-1	Dt/S-1 (5 wt%)
*S_BET_* (m^2^/g)	10.0	532.0	398.8
*V_micropore_* (cm^3^/g)	0.005	0.170	0.131
*V_total_* (cm^3^/g)	0.045	0.515	0.342

**Table 2 materials-12-00551-t002:** Equilibrium adsorption capacities (mmol/g) of three probe volatile organic compounds (VOCs) on various adsorbents.

Dt/S-1 (wt%)					S-1	Dt
Sample	1	5	10	20		
Acetone	0.82 ± 0.04	1.01 ± 0.06	1.02 ± 0.05	0.80 ± 0.04	1.01 ± 0.05	0.07 ± 0.002
Ethyl acetate	1.10 ± 0.05	1.31 ± 0.08	1.28 ± 0.08	0.92 ± 0.04	1.28 ± 0.08	0.06 ± 0.002
Toluene	0.62 ± 0.02	0.71 ± 0.03	0.69 ± 0.02	0.48 ± 0.01	0.72 ± 0.03	0.12 ± 0.005

**Table 3 materials-12-00551-t003:** Equilibrium adsorption capacities (mmol/g) of Dt/S-1 (5 wt%) prepared at different temperatures.

Sample	90 °C	100 °C	110 °C
Acetone	0.58 ± 0.02	1.01 ± 0.06	0.97 ± 0.05
Ethyl acetate	0.90 ± 0.05	1.31 ± 0.08	1.22 ± 0.07
Toluene	0.48 ± 0.02	0.71 ± 0.04	0.68 ± 0.03

**Table 4 materials-12-00551-t004:** Equilibrium adsorption capacities (mmol/g) of Dt/S-1 (5 wt%) prepared at different times.

Sample	3 Days	4 Days	5 Days
Acetone	0.88 ± 0.05	1.01 ± 0.06	0.98 ± 0.05
Ethyl acetate	1.17 ± 0.06	1.31 ± 0.08	1.19 ± 0.07
Toluene	0.71 ± 0.04	0.71 ± 0.03	0.69 ± 0.04

**Table 5 materials-12-00551-t005:** Adsorption kinetics parameters of three probe VOCs. *Q_e_* = adsorption capacity at equilibrium. *k_1_* = pseudo first-order rate constant of adsorption. *k_2_* = pseudo second-order rate constant of adsorption. *R*^2^ = correlation coefficient.

Model	Pseudo First-Order Model				Pseudo Second-Order Model		
Parameter	*Q_e_* (exp, mmol/g)	*Q_e_* (cal, mmol/g)	*k_1_*	*R* ^2^	*Q_e_*(cal, mmol/g)	*k_2_*	*R* ^2^
Acetone	0.99	0.91	0.21	0.9874	1.04	0.27	0.9981
Ethyl acetate	1.31	1.22	0.17	0.9908	1.39	0.17	0.9995
Toluene	0.69	0.62	0.22	0.9948	0.75	0.35	0.9993

**Table 6 materials-12-00551-t006:** Adsorption isotherm parameters of three probe VOCs. *Q_max_* (mol·g^−1^) = maximum adsorption capacity. *k_H_* (L·g^−1^) = Henry constant. *k_L_* (L·mmol^−1^) = Langmuir equilibrium constant. *k_F_* (mmol^1−n^·L^n^·g^−1^) = Freundlich constant. 1/n = heterogeneity factor.

Model	Henry		Langmuir			Freundhch		
Parameter	*k_H_*	*R* ^2^	*k_L_*	*Q_max_*	*R* ^2^	*k_F_*	*n*	*R* ^2^
Acetone	9.84	0.8250	13.24	1.85	0.9974	3.50	0.53	0.9933
Ethyl acetate	19.15	0.7028	32.45	1.93	0.9986	4.62	0.47	0.9787
Toluene	8.08	0.8601	12.07	1.41	0.9903	3.03	0.60	0.9743

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
