# Peer review of "Fabrication of Diatomite/Silicalite-1 Composites and Their Property for VOCs Adsorption"

_materials, 2019, doi:10.3390/ma12040551_

Round 1
Reviewer 1 Report
After reading the manuscript entitled:
"Fabrication of Diatomite/Silicalite-1 composites and their property for VOCs adsorption" I found the title attractive to wide researchers interested in this area. However, I have the following comments to address: 1-Ithenticate test of the manuscript indicated 39% with references and 26% without references and this should not exceed 10%. 2-In the abstracte (page 1, line, 17) The authors mentioned that effect of zeolite content and hydrothermal temperature were studied. While the work and characterization was performed by one sample (5%) for characterization, kinetics and equilibrium. only adsorption capacity was measured for all prepared samples. 3-Figure 2 in page 4, the magnification bar must be clear. 4- In Table 1, BET analysis was performed only for one sample (5% at one temperature) and this is not enough to get clear idea as it is very well known that the amount of functional group added to a support strongly affect its morphological structure including surface area, pore voluve and pore size. Furthermore, this analysis must be performed for samples prepared under different temperature since temperature also has strong effect. 5-Figure 3 should show the isotherms of all prepared samples to compare between them. 6-In Page 5, line 198, it was mrntioned that adsorption capacities increased by increasing the amount of S-1 added up to 20% then started to decrease. this can be explained by results of BET analysis of all samples to show the effect on surface area, pore size , and pore volume of the samples. 7-In page 5, line 203, it was mentioned that incresing S-1 leads to agglomeration and pore blockage, again BET analysis is important to proof this. 8-In page 6, line 229, it was mentioned that that crystallinty improved by increasing temperature. This statement should be proofed by XRD analysis of different samples. 9- Kinetics analysis and equilibrium analysis was perforemed only by one sample, why? 10- No need for Table 7, simply write the values in the discussion. 11- Again in the conclusion, it was mentioned that the effect of doping ratio and hydrothermal temperature effects were investigated, while only adsorption capacity was studied while crystalinity and surface morphology were not investigated. Also kinetics and equilibrium was performed with a single sample.To approve a single suggestion, mouse over it and click "✔"Click the bubble to approve all of its suggestions.
Author Response
Dear Reviewer,
We have studied the valuable comments from you carefully, and tried our best to revise the manuscript. The point to point responds are listed as following:
Point 1: Ithenticate test of the manuscript indicated 39% with references and 26% without references and this should not exceed 10%.
Response 1: We have careful recheck our manuscript and modified it.
Point 2: In the abstracte (page 1, line, 17) The authors mentioned that effect of zeolite content and hydrothermal temperature were studied. While the work and characterization was performed by one sample (5%) for characterization, kinetics and equilibrium. only adsorption capacity was measured for all prepared samples.
Response 2: Thank you for your valuable advice. We have added some characterizations of other samples in the Supporting Information.
Point 3: Figure 2 in page 4, the magnification bar must be clear.
Response 3: Thank you for your careful work. The scale has been redrawn.
Point 4: In Table 1, BET analysis was performed only for one sample (5% at one temperature) and this is not enough to get clear idea as it is very well known that the amount of functional group added to a support strongly affect its morphological structure including surface area, pore voluve and pore size. Furthermore, this analysis must be performed for samples prepared under different temperature since temperature also has strong effect.
Response 4: Thank you for your valuable and thoughtful comments. The surface area, pore volume and size of the other samples prepared in different conditions were introduced in Table S1 in the Supporting Information and described in the manuscript.
Point 5: Figure 3 should show the isotherms of all prepared samples to compare between them.
Response 5: Thank you very much. The isotherm of S-1 was added into the Fig. 3, and the isotherms of other samples was added in the Fig. S1.
Point 6: In Page 5, line 198, it was mrntioned that adsorption capacities increased by increasing the amount of S-1 added up to 20% then started to decrease. this can be explained by results of BET analysis of all samples to show the effect on surface area, pore size, and pore volume of the samples.
Response 6: Thank you for your instructive suggestions. The BET analysis was added in Table S1 and Fig. S1. We had discussed the results as follow: “In addition, with the increase of seeds contents from 1wt% to 10wt%, ethyl acetate adsorption capacities on the adsorbents were improved from 1.10 to 1.31 mmol/g, as well as the specific surface area of the samples increased from 319.2 m2/g to 402.3 m2/g (Table S1). However, the adsorption capacities of three VOCs declined obviously as the seeds contents up to 20wt%. The specific surface area of Dt/S-1(20wt%) decreased to 336.8 m2/g. While the micropore volume improved, the total volume had decreased, indicating that the dispersed S-1 nanocrystals were occupied inside the pores of Dt.”
Point 7: In page 5, line 203, it was mentioned that incresing S-1 leads to agglomeration and pore blockage, again BET analysis is important to proof this.
Response 7: Thank you for your valuable advice. The BET analysis was added in Table S1 and Fig. S1.
Point 8: In page 6, line 229, it was mentioned that that crystallinty improved by increasing temperature. This statement should be proofed by XRD analysis of different samples.
Response 8: Thank you very much. The XRD analysis was added as Fig. S2.
Point 9: Kinetics analysis and equilibrium analysis was perforemed only by one sample, why?
Response 9: Thank you for your valuable and thoughtful comments. In our research we found that the Dt/S-1(5wt%) and Dt/S-1(10wt%) had the best adsorption effect on probe VOCs, but considering the economy of synthesis, Dt/S-1(5wt%) was better than Dt/S-1(10wt%). Then we did an in-depth analysis for Dt/S-1(5wt%).
Point 10: No need for Table 7, simply write the values in the discussion.
Response 10: Thank you very much. According to your comments, Table 7 was deleted and described as follow “The calculated kd of acetone, ethyl acetate and toluene were 6.25, 15 and 0.47, and the α values of ethyl acetate to acetone and toluene were 2.4 and 31.91, respectively.”
Point 11: Again in the conclusion, it was mentioned that the effect of doping ratio and hydrothermal temperature effects were investigated, while only adsorption capacity was studied while crystalinity and surface morphology were not investigated. Also kinetics and equilibrium was performed with a single sample.
Response 11: Thank you for your careful reading of our manuscript. We acknowledge that the previous statement was inaccurate and we have modified it as follow: “The Dt/S-1(5wt%) exhibited a considerably higher VOCs adsorption capacity compared to raw Dt and other composites prepared under different conditions.”

Reviewer 2 Report
The paper presents the way of preparation of Diatomite/Silicalite-1 composites and their application in the VOCs adsorption. The presented results are very interesting, however, in my opinion, some statements should be clarified:
1. The authors state in the paper, that the effect of seeded zeolite contents (1wt%, 5wt%, 10wt%, 20wt%), hydrothermal temperature (90oC, 100oC, 110oC) and hydrothermal time (3 days, 4 days, 5 days) were investigated. However, analyzing the results, only one composite (Dt/S-1(5wt%)) is described in detail. Are there any research results for the other composites?
2. In my opinion, the structural parameters (SBET and Vp) of these materials (synthesized under different conditions) should be also discussed in the manuscript. These parameters have a significant impact on the adsorption volume.
3. page 1, line 10 “This work prepared hierarchical porous materials using…” - incorrect statement
4. page 1, line 10 “supporter” – should be “support”?
5. page 1 line 15: The adsorption capacities of Dt/S-1 composites for three probe VOCs (ethyl acetate, acetone, and toluene) were rather superior to Dt, and the adsorption capacity for ethyl acetate > acetone> toluene” – not clear statement
6. page 2 line 58: “SBET” - use the subscript
7. page 3 line 124: “N2 adsorption-desorption isotherms …” - conditions should be added
8. page 3 line 125: “multi-point Brunauer–Emmett–Teller (BET)… “ - reference should be added
9. page 3 line 132: “with a flame ionization detector…” - please indicate the type of stationary phase and process conditions
10. page 4 line 148: “Steamed Bun-shaped” – what does it mean?
11. line 162: “with respect to the N2 activation procedure…” - N2 adsorption is not an activation procedure but the way of determination of structural parameters of adsorbent. Please explain it.
12. line 167: Figure 2: the scale of SEM images is rather poor quality
13. line 174: Table 1: “Vmicropore (m3/g) and Vtotal (m3/g)” Is this unit (m3/g) correct?
14. line 226: “Table 3 introduced the equilibrium…” – incorrect statement
15. line 228: “With the increase of temperature, the crystallinity was improved accordingly.” - Do the authors have data to confirm crystallinity improvement? Why does the structure of the composite change due to the hydrothermal modification?- please explain
16. line 232: “Table 4 introduced the equilibrium…” incorrect statement
17. line 273: Table 6: Enter the appropriate subscripts and superscripts
18. line 280: ” Later, with the increase of adsorption to ethyl acetate, the adsorption to acetone decreased rapidly, since ethyl acetate began to seize the adsorption sites.” The statement is linguistically incorrect
19. line 298: Table 7: Enter the appropriate subscripts
Author Response
Dear Reviewer,
We have studied the valuable comments from you carefully, and tried our best to revise the manuscript. The point to point responds are listed as following:
Point 1: The authors state in the paper, that the effect of seeded zeolite contents (1wt%, 5wt%, 10wt%, 20wt%), hydrothermal temperature (90oC, 100oC, 110oC) and hydrothermal time (3 days, 4 days, 5 days) were investigated. However, analyzing the results, only one composite (Dt/S-1(5wt%)) is described in detail. Are there any research results for the other composites?
Response 1: Thank you very much. In our research results, the composite (Dt/S-1(5wt%)) has the largest equilibrium adsorption capacities, so we have focused on it and investigated more. We have described some research results for other composites in detail in the supporting information this time.
Point 2: In my opinion, the structural parameters (SBET and Vp) of these materials (synthesized under different conditions) should be also discussed in the manuscript. These parameters have a significant impact on the adsorption volume.
Response 2: Thank you for your valuable advice. The surface area, pore volume and size of the other samples prepared in different conditions were introduced in Table S1 and Fig. S1 in the Supporting Information and described in the manuscript.
Point 3: page 1, line 10 “This work prepared hierarchical porous materials using…” - incorrect statement
Response 3: We are sorry for this language mistake. It was corrected as “In this work we prepared hierarchical porous materials using…”
Point 4: page 1, line 10 “supporter” – should be “support”?
Response 4: Thank you for your careful reading of our manuscript. The word “supporter” was replaced by “support”.
Point 5: page 1 line 15: The adsorption capacities of Dt/S-1 composites for three probe VOCs (ethyl acetate, acetone, and toluene) were rather superior to Dt, and the adsorption capacity for ethyl acetate > acetone> toluene” – not clear statement
Response 5: Thank you very much. We have modified it as follow: “ The adsorption capacities of Dt/S-1 composites for three probe VOCs (ethyl acetate, acetone, and toluene) were rather superior to Dt, and the composites had preferential adsorption selectivity for ethyl acetate.”
Point 6: page 2 line 58: “SBET” - use the subscript
Response 6: Thank you for your careful reading of our manuscript. It has been corrected.
Point 7: page 3 line 124: “N2 adsorption-desorption isotherms …” - conditions should be added
Response 7: Thank you for your valuable advice. The conditions were added as follow: “A Micromeritics ASAP 2020 system was used to measure the N2 adsorption-desorption isotherms (N2 adsorption isotherms at 77K, and all samples were pre-activated at 300 oC under vacuum for 10h)”
Point 8: page 3 line 125: “multi-point Brunauer–Emmett–Teller (BET)… “ - reference should be added
Response 8: Thank you very much. The reference has been added as “multi-point Brunauer–Emmett–Teller (BET) equation [24]”
Point 9: page 3 line 132: “with a flame ionization detector…” - please indicate the type of stationary phase and process conditions
Response 9: Thank you for your instructive suggestions. According to your comments, the stationary phase and process conditions were indicated as follow: “The concentrations of residual organics were determined by a gas chromatograph device (GC-2014C, SHIMADZU, Japan) equipped with a WonderCap5 column and a flame ionization detector (FID). The temperatures of inlet, analyzer and column were 240 oC, 300 oC and 70 oC, respectively. The equilibrium time was 4 min.”
Point 10: page 4 line 148: “Steamed Bun-shaped” – what does it mean?
Response 10: Thank you very much. This is an incorrect statement and we have removed it.
Point 11: line 162: “with respect to the N2 activation procedure…” - N2 adsorption is not an activation procedure but the way of determination of structural parameters of adsorbent. Please explain it.
Response 11: Thank you for your careful reading of our manuscript. We acknowledge that the previous statement was inaccurate and we have modified it as follow: “The surface properties of Dt and Dt/S-1(5wt%) determined by the N2 adsorption-desorption…”
Point 12: line 167: Figure 2: the scale of SEM images is rather poor quality
Response 12: Thank you for your careful work. The scale has been redrawn.
Point 13: line 174: Table 1: “Vmicropore (m3/g) and Vtotal (m3/g)” Is this unit (m3/g) correct?
Response 13: Thank you for your careful work. It is not correct, we’ve corrected it to be “cm3/g”
Point 14: line 226: “Table 3 introduced the equilibrium…” – incorrect statement
Response 14: Thank you very much. We modified it as follow: “The equilibrium adsorption capacities of Dt/S-1 (5wt%) prepared at different hydrothermal temperature are shown in Table 3”
Point 15: line 228: “With the increase of temperature, the crystallinity was improved accordingly.” - Do the authors have data to confirm crystallinity improvement? Why does the structure of the composite change due to the hydrothermal modification?- please explain
Response 15: Thank you for your valuable advice. According to your comments, we added the X-ray diffraction (XRD) patterns of the Dt/S-1(5wt%) synthesized at different temperature (Figure S2) in the supporting information. It can be clearly seen from the XRD patterns that the crystallinity of the samples increases with the increase of temperature.
Point 16: line 232: “Table 4 introduced the equilibrium…” incorrect statement
Response 16: Thank you very much. It was modified to “Table 4 shows the equilibrium adsorption capacities of Dt/S-1 (5wt%) prepared at different hydrothermal time.”
Point 17: line 273: Table 6: Enter the appropriate subscripts and superscripts
Response 17: Thank you for your careful work. It was corrected.
Point 18: line 280: ” Later, with the increase of adsorption to ethyl acetate, the adsorption to acetone decreased rapidly, since ethyl acetate began to seize the adsorption sites.” The statement is linguistically incorrect
Response 18: Thank you for your careful work. We modified it as follow: “Subsequently, the adsorption of ethyl acetate continued to increase, while the adsorption of acetone decreased sharply.”
Point 19: line 298: Table 7: Enter the appropriate subscripts
Response 19: Thank you for your careful work. Table 7 was deleted and described as follow “The calculated kd of acetone, ethyl acetate and toluene were 6.25, 15 and 0.47, and the α values of ethyl acetate to acetone and toluene were 2.4 and 31.91, respectively.” according to the other reviewer.

Round 2
Reviewer 2 Report
In my opinion, the authors in the revised version of the manuscript took into account all my comments.